# Exploratory Study to Evaluate the Impact of Interim PET/CT Assessment in First-Line Follicular Lymphoma

**DOI:** 10.3390/cancers17071065

**Published:** 2025-03-21

**Authors:** María Poza, Alejandro Martin-Muñoz, Patricia López-Pereira, Gloria Figaredo, Irene Zamanillo, Rodrigo Íñiguez, Ana Carla Oliveira, Tycho Baumann, Antonia Rodríguez-Izquierdo, Carlos Grande, Pilar Sarandeses, Enrique Revilla, Montserrat Cortés, Rosa Ayala, María Calbacho, Joaquín Martínez, Santiago Barrio, Ana Jiménez-Ubieto

**Affiliations:** 1Department of Hematology, Hospital Universitario 12 de Octubre, Instituto de Investigación Sanitaria Hospital 12 de Octubre (imas12), CNIO, CIBERONC, 28041 Madrid, Spainrodriiguez2293@gmail.com (R.Í.); tycho.baumann@gmail.com (T.B.); jmarti01@med.ucm.es (J.M.); 2Altum Sequencing Co., 28005 Madrid, Spain; 3ICO (Instituto Catalán de Oncología) L’Hospitalet, 08908 Bacelona, Spain; patricialopez@iconcologia.net (P.L.-P.); acoliveira@iconcologia.net (A.C.O.); mcortes@bellvitgehospital.cat (M.C.); 4Department of Hematology, Hospital Gregorio Marañón, 28007 Madrid, Spain; 5Department of Hematology, Clínica Universitaria de Navarra, 31008 Madrid, Spain; cgrandeg@unav.es; 6Nuclear Medicine Department, Hospital 12 de Octubre, 28041 Madrid, Spain; mariadelpilar.sarandeses@salud.madrid.org; 7Pathology Department, Hospital 12 de Octubre, 28041 Madrid, Spain; 8Lymphoma Research Group, Department of Medical Oncology, Hospital Universitario Puerta de Hierro-Majadahonda, IDIPHISA, 28222 Madrid, Spain

**Keywords:** follicular lymphoma 2, PET/CT, interim evaluation

## Abstract

This study investigates how interim PET (iPET) scans performed during treatment can help predict outcomes for patients with follicular lymphoma undergoing their first treatment. By analyzing data from 117 patients, the study found that those with positive iPET results, defined by a Deauville score of 4 or 5, had significantly worse progression-free survival (PFS) compared to negative iPET patients. The five-year PFS rate was 34% for iPET-positive patients versus 76% for iPET-negative patients. These findings highlight iPET as an independent predictor of relapse risk, suggesting its potential role in response-adapted treatment strategies.

## 1. Introduction

Follicular lymphoma (FL) is considered an indolent disorder with a relatively favorable outcome [1,2]. It is the second most common non-Hodgkin lymphoma (NHL) in developed countries, accounting for approximately 20–25% of NHL worldwide [3,4]. Several immunochemotherapeutic (ICT) combinations or a regimen of lenalidomide and rituximab remain valid options for many patients [5,6,7,8,9]. Rituximab maintenance as a post-induction consolidation strategy is well established by the phase III PRIMA and GALLIUM studies [8,10]. Most patients achieve long-term remissions, with median survival rates of almost 20 years [8,10]. However, while the main aims of achieving remission and improving quality of life have been met, there are several other unmet needs: first, to improve survival in the 15–20% of patients who are unreactive to initial treatment or who experience disease progression during the first two years after first-line therapy (POD24) [11,12,13]; second, to mitigate the toxicity of anti-CD20 maintenance for patients who achieve full metabolic remission [14].

In order to address these unmet needs, various clinical, molecular, pathological, and imaging biomarkers have been shown to predict survival rates at the time of diagnosis, and could be used to determine the optimal therapy. However, many of these tools are inaccessible in daily clinical practice and have not been sufficiently tested [3,4,5,6,7,8,9,10,15,16,17,18,19,20,21,22,23]. Several studies confirm that the results of end-of-induction (EOT) positron emission tomography (PET) after ICT strongly predict progression-free survival (PFS) and overall survival (OS) [24,25]. Likewise, EOT minimal residual disease (MRD) has an impact on prognostics [26]. In cases of an absence of OS benefits and increased risk of infection with frontline antibody maintenance, using PET might be relevant to guide therapeutic approaches, even more so in the context of the COVID-19 pandemic and high infection rates [8,10,14,27]. Dynamic risk assessment using EOT PET [14,28] or EOT PET plus MRD measured via Polymerase Chain Reaction (PCR) of the t(14;18) translocation is currently under evaluation as a response-adapted treatment platform. Nevertheless, when using PET and/or MRD at EOT, patients have already undergone the complete cycle of ICT. If the results from interim PET were equally predictive, they would allow for the earlier identification of patients with a high risk of poor outcomes, and could facilitate anticipated interventions (including potentially earlier changes in therapy).

In contrast to classic Hodgkin’s lymphoma or diffuse large B-cell lymphoma (DLBCL) [29,30,31,32], the prognostic value of iPET results in FL remains uncertain. While some studies suggest that positive iPET results are associated with poorer PFS [25,33,34], others indicate that they have no prognostic significance [35,36].

The primary aim of this study was to determine whether iPET could identify FL patients at high risk of relapse after receiving systemic frontline therapy (mainly ICT), and to establish a proof-of-concept platform for clinical trials designed to evaluate response-adapted treatment following only four cycles of initial ICT.

## 2. Materials and Methods

### 2.1. Patients

We retrospectively evaluated newly diagnosed FL patients who received first-line therapy and underwent basal, interim (after four cycles of therapy), and EOT PET scans between 2012 and 2022 at three academic hospitals from the GELTAMO (Grupo Español de Linfomas/Trasplante Autólogo de Médula Ósea) group (Hospital Universitario 12 de Octubre, Instituto Catalán de Oncología and Complejo Hospitalario de Toledo). The inclusion criteria were the following: (1) being over 18 years; (2) having a histologically confirmed FL grade 1–2 or 3A; (3) fulfilling the GELF (Groupe D’Etude des Lymphomes Folliculaires) criteria at the moment of initiating systemic treatment; and (4) having an iPET final PET scan. We obtained clinical, laboratory, treatment, and PET examination data (Figure 1).

### 2.2. Analysis of PET/CT Imaging

All patients were required to fast for at least 6 h prior to the PET scan. Basal, interim, and EOT PET scans were performed by expert nuclear medicine radiologists who were blind to patient outcomes. PET examinations were performed using either the General Electric Discovery MI (GEDMI) scanner or the Siemens Biograph 6 scanner, following injection of 2.5–5 MBq/kg 2-deoxy-2-[fluorine-18]fluoro-D-glucose (FDG). The calibration of the machines is performed annually at the Hospital 12 de Octubre, and they undergo yearly AERL accreditation from the European Association of Nuclear Medicine (EANM). PET scans obtained with a low-dose protocol were used for anatomic localization and attenuation correction of the PET images. PET scans were scored by experienced PET physicians using the Deauville score (DS) standards in agreement with Lugano classification definitions: (1) no uptake above background; (2) uptake ≤ mediastinum; (3) uptake > mediastinum but ≤ liver; (4) uptake > liver; and (5) uptake markedly > liver or area of new disease [37]. A positive PET result was defined by a DS of 4 or 5, while scores from 1 to 3 were considered negative or complete response (CR). The maximum standardized uptake value (SUVmax) was recorded during both the basal and interim PET scans. In cases where PET scans did not show FDG uptake, the SUVmax was recorded as 1. We calculated the ΔSUVmax by dividing the difference in the SUVmax between the basal PET and interim PET by the basal SUVmax. When the basal SUVmax was inferior to 10, we considered it not assessable [38]. Histological transformation (HT) into high-grade B-cell lymphoma was suspected due to imaging or clinical changes and was confirmed by performing a tissue biopsy.

### 2.3. Statistics

All statistical analyses were performed using SPSS^®^ Statistics, V21.0 (SPSS Inc., Chicago, IL, USA). The primary endpoint was PFS, measured from the date of treatment initiation until the date of disease progression, death from any cause, or last follow-up. Disease progression was determined through an imaging evaluation and suspicion of HT was confirmed or ruled out by biopsy. OS was defined as the time from the initiation of treatment to death from any cause or last follow-up. Univariate and multivariate Cox regressions were used to evaluate links between demographic, clinical, and analytical prognostic factors and both PFS or OS, with hazard ratios (HR) and 95% confidence intervals (CI). The value of the most important prognostic scores (FLIPI, FLIPI2, and PRIMA-PI) was also analyzed. iPET and EOT PET values were also determined via Cox regression and Kaplan–Meier curves. All statistical tests were two-sided and a cut-off *p* value of <0.05 was considered statistically significant.

## 3. Results

### 3.1. Patient Characteristics

The baseline characteristics of the 121 patients are shown in Table 1. The median age at diagnosis was 62 years (range 27–87), 52% of patients were male, and 75% of patients had FL grade 1–2. Most patients (92.5%) had advanced Ann Arbor stage and only 24% had B symptoms at the time of diagnosis. High-risk FLIPI, FLIPI2, and PRIMA-PI scores were observed for 58%, 46%, and 31.5% of patients, respectively. Patients received treatment at a median of one month (range 0–94) after diagnosis.

Only 16 patients (14%) were initially assigned to a watch-and-wait strategy. All of them were finally treated because of meeting the GELF criteria. Consistently with current management strategies for FL (6,9,11), 77% of patients (n = 90) received R-CHOP (rituximab, cyclophosphamide, doxorubicin, vincristine, and prednisone) treatment, whereas 21% (n = 24) underwent RB (rituximab and bendamustine) treatment. The remaining two patients received anti-CD20 or duvelisib. Of the 118 patients receiving ICT, 114 (97%) completed 6 cycles. Most patients (87%) underwent rituximab maintenance, which was not administered to 15 patients because of adverse events (n = 6), physician decisions (n = 2), intercurrent malignancy (n = 2), or other unknown reasons (n = 5).

The median follow-up period for all patients was 34 months (range 3–115). During the follow-up period, 19 patients (15.7%) experienced POD24 after ICT, whereas 13 patients (10.7%) relapsed more than 24 months after receiving ICT. Twelve patients died, with three deaths being attributed to lymphoma progression, two to secondary neoplasia, and seven to infectious complications. The 5-year PFS and OS rates were 64% (95% CI 58–69%) and 88% (95% CI 84–92%), respectively. During follow-up, three patients showed HT into high-grade B-cell lymphoma at 6, 10, and 72 months.

### 3.2. Risk Factors for Progression-Free Survival Analysis

Univariate analysis revealed that the following factors were related to an inferior PFS rate: more than four nodal areas, an elevated β2 microglobulin, hemoglobin levels below 12 g/dL, elevated serum lactate dehydrogenase (LDH), and not receiving rituximab maintenance (Table 2). A prior watch-and-wait strategy or type-ICT regimen (rituximab and bendamustine vs. R-CHOP) did not predict worse outcomes. None of the prognostic factors analyzed demonstrated a prognostic impact on OS rates.

We also analyzed the prognostic implications of FLIPI, FLIPI2, and PRIMA-PI scores (Figure 2). All three indexes were prognostic of PFS outcomes. FLIPI had the highest sensitivity index (82%) in detecting a relapse, and its positive predictive value was 41%. The prognostic impact of these scores was not evidenced in patients who received RB treatment, probably because of their limited number.

### 3.3. Prognostic Value of iPET

The Deauville score (DS) was defined for 113 patients with Interim PET/CT scans (iPET), with the remaining 4 only being evaluated via CT (all in complete response). A total of 10 patients (9%) were classified as DS1, 26 (23%) as DS2, 40 (35%) as DS3, 30 (27%) as DS4, and 7 (6%) as DS5. The PFS curves overlapped for patients with an iPET DS of 1, 2, or 3, as well as for patients with a DS of 4 or 5 (Figure 3A). A similar result was observed for EOT PET (Appendix A). Thus, we continued to categorize DS3 as iPET (−). The patients with DS5 did not show a higher rate of POD24 than patients with DS4 (2-year PFS rate of 44.4% vs. 54.4%, *p* = 0.767).

Overall, 40 patients (34%) were considered iPET (+). The estimated 5-year PFS rate was 76% for patients with a negative iPET in comparison with 34% for those with a positive iPET (HR 4.3, 95% CI 2.1–8.8; *p* < 0.001) (Figure 3B). No significant difference in OS was observed (*p* = 0.599, Appendix A). A percentage of 34% and 7.6% of patients with iPET (+) and (−) experienced POD24, respectively (HR 5.6, 95% CI 2.1–14.6; *p* < 0.001). Negative iPET scores showed a negative predictive value of 94% for POD24.

Basal SUVmax was obtained for 111 patients (95%). with a median of 12.7 (range 4–57.2). Interim SUVmax was recorded for 94 patients (80%), with a median of 3.1 (range 1–37.9); of those, 33 (36%) showed values greater than 3.7. Additionally, we could calculate the ΔSUVmax in 57 patients. Interim ∆SUVmax was not a significant predictor of PFS (*p* = 0.154). Patients with a ∆SUVmax of less than 71% had a 5-year PFS of 51% (95% CI 53–86%) in comparison with 75% (95% CI 70–95%) for those with a ∆SUVmax greater or equal to 71% (Appendix A).

In a multivariate analysis including variables associated with a lower PFS rate (nodal areas > 4, elevated β2 microglobulin, hemoglobin < 12 g/dL, elevated LDH, rituximab maintenance, high-risk FLIPI, and high-risk PRIMA-PI), an iPET (+) and high-risk FLIPI were significant predictors of PFS (HR 4.2, 95% CI 1.7–10.3, *p* = 0.022 and HR 3.7, 95% CI 1.2–11.0, *p* = 0.021, respectively).

### 3.4. Prognostic Value of EOT PET

An EOT PET/CT scan was assessable for 114 patients (3 patients were evaluated via CT, all in complete response). A DS of 1, 2, 3, 4, and 5 was assigned to 27 patients (23%), 26 patients (22%), 35 patients (30%), 17 patients (15%), and 9 patients (8%), respectively (Appendix A). Overall, 26 patients (23%) were considered EOT PET (+). The EOT CR rate was 81.5%. The EOT PET findings were predictive of relapse, as patients with EOT PET (+) had a 2-year PFS of 48.5% in comparison with 90.9% of those with EOT PET (−) (HR 5.1, 95% CI 2.5–10.4, *p* < 0.001) (Appendix A). DS1 was more frequently observed in EOT PET than in iPET, while DS4 was less commonly reported in EOT PET. Among the patients with EOT PET (+), 60% (15 out of 25) relapsed in a median of 7 months (range 4–55); among patients with EOT PET (−), 18.5% (17 out of 92) relapsed in a median of 26 months (range 10–88). We did not observe a significant difference in OS between patients with EOT PET (+) and EOT PET (−). A total of 48% and 7.7% of patients with EOT PET (+) and (−) presented POD24, respectively (HR 11.1, 95% CI 3.7–33.3, *p* < 0.001). As previously described for iPET, PFS curves overlapped for patients with an EOT PET DS of 1, 2, or 3, and DS 4 and 5 (Appendix A).

In a multivariate analysis including variables associated with a lower PFS rate, we obtained similar results to those for iPET: EOT PET and high-risk FLIPI remained significant predictors of PFS (HR 4.6, 95% CI 1.5–13.6, *p* = 0.007 and HR 3.5, 95% CI 1.2–10.6, *p* = 0.026).

### 3.5. Dynamics of PET Analysis

Patients who converted from a positive iPET to a negative EOT PET (n = 20) had inferior PFS to those who had both examinations negative (n = 75, HR 3.9, CI 95% 1.14–10.6, *p* = 0.012). Also, we observed no statistical differences between patients with both examinations positive (n = 18) and patients with a positive iPET and negative EOT PET (n = 20, HR 1.9, CI 95% 0.71–5.1, *p* = 0.184) (Figure 4). Conversion from a negative iPET to a positive EOT PET was very rare (four patients), two of which showed disease progression at 5 and 13 months. After more than 4 years of follow-up, the other two remained progression-free.

## 4. Discussion

This retrospective study is one of the largest examining the prognostic value of iPET and EOT PET scans in FL frontline therapy. Additionally, we analyzed the prognostic significance of other clinical and analytical risk factors associated with disease progression. Consistent with the current management strategies for FL [6,9,10], most patients received treatment with R-CHOP and underwent rituximab maintenance. The fraction of patients that experienced POD24 was similar to the previously reported rates [12,13].

The PET scan responses assessed using DS, SUVmax, and ∆SUVmax were similar to those previously described [25,34]. The iPET was a strong predictor of PFS, with supporting data recently published by Merryman et al., who proved iPET to be a significant predictor of PFS in FL patients [34]. In the multivariate analysis, only iPET (+) and high-risk FLIPI were significant for PFS, indicating that a dynamic evaluation of responses could better identify patients who require early intervention [39,40]. Interestingly, patients with a DS of 4–5 on an iPET scan (observed in 34% of FL grade 1–3A patients) who later showed a negative EOT PET had inferior PFS to those for whom both scans were negative. These findings suggest that iPET imaging could help identify patients with a negative EOT PET who are at high risk of relapse. With this information and taking into account the recent approval of highly effective immunotherapy strategies in FL [41,42,43], we consider that in the near future, interim PET could be used to change treatment earlier. In contrast, only a small fraction of patients with a DS of 1–3 on an iPET scan had POD24, in comparison with the patients with a DS of 4–5. Although it should be taken with care, the lower POD24 rates observed in the patients under rituximab maintenance in the iPET (+) subgroup is of particular interest. These results, together with the extremely positive outcomes seen in patients with low-risk FLIPI, are enough to safely forego EOT PET in patients with DS1–3, plus low FLIPI risk at diagnosis. Would it be safe to forego EOT PET with DS 1–3 and low FLIPI? More studies and a larger number of patients are needed for us to be able to recommend this in high-risk FLIPI, FLIPI-2, or PRIMA-IP patients. 

PET has a crucial role in initial staging and EOT response assessment in FL [37]. EOT PET has been proven to be highly predictive of prognosis within FL patients groups [24,25]. However, few studies have evaluated the prognostic value of iPET scans [25,33,34,35,36]. Nearly all these studies used older PET response criteria (e.g., International Harmonization Project criteria) and were limited by small sample sizes. More recently, Merryman et al. investigated the use of the DS for the interpretation of iPET scans in a retrospective cohort of 128 patients with FL grade 1–3B [34]. Interim PET scans were performed after 2–4 cycles of ICT and proved to be a significant predictor of PFS. Patients who had a DS of 4–5 had worse PFS rates than those who had a DS of 1–2 [34].

Developing response-adapted clinical trials on the basis of only PET scans has some limitations, such as the tests’ limited sensitivity and specificity and the subjective radiologist’s interpretation of the results [44,45]. Recently, our study group and Fernández-Miranda et al. showed that cfDNA MRD positivity in liquid biopsy during an interim evaluation had a prognostic impact on PFS in a small group of patients treated with first-line ICT [37,44]. The combination of cfDNA MRD could be complementary to iPET results for establishing a prognosis and adapting the therapy.

From a translational/clinical perspective, these findings indicate that the use of PET for the evaluation of responses in FL has the potential to serve as a tool for designing response-adapted treatment strategies. In recent years, the clinical need for response-adapted maintenance led to the use of EOT PET for guiding treatment in three different clinical trials [28,39,46]. In the FOLL12 clinical trial, 807 patients were randomly assigned to receive response-adapted ICT treatment guided by PET and molecular studies. The authors hypothesized that intensifying interventions for patients with the highest putative risk of poor survival, defined as those with persistent evidence of illness on the basis of molecular evaluation (MRD-positive) and metabolic imaging (PET-positive), would yield the greatest impact on outcomes. Despite its unfavorable global results [39], this study demonstrates the feasibility of a response-adapted approach in FL using EOT PET scans.

Considering our findings, iPET evaluation should be considered a valuable biomarker for FL patients receiving frontline ICT. It could also be used to define treatment-adapted clinical trials. The earliest-possible use of immune-based therapies (e.g., CAR T cell therapy and bispecific antibodies) could be a promising alternative for iPET (+) patients who are likely to have poor outcomes with ICT treatments.

To our knowledge, this is the second study exploring the significance of ΔSUVmax in the iPET scans of first-line FL patients. In contrast to Merryman’s findings [34], we cannot strongly propose an optimal ΔSUVmax threshold for further investigation in prospective clinical trials.

Our study has some limitations, such as the small sample size and the limited follow-up in indolent diseases. Nevertheless, 97.5% of the patients received ICT, leading to a homogeneous cohort of 118 patients. The main inconvenience of the study is the absence of a centralized nuclear medicine radiologist to review all the PET scans. Nevertheless, most patients underwent evaluation on nearly two occasions to avoid potential bias. We recognize other important sources of variability among our study cohorts, such as differences in the generation of the PET scanners utilized. However, it is worth noting that all patients underwent iPET assessments at the same time.

## 5. Conclusions

In conclusion, the data suggest that a positive iPET scan can predict PFS and POD24 in first-line FL patients. iPET positivity (Deauville score 4–5) was strongly associated with a higher risk of disease progression and poorer progression-free survival (PFS), while iPET negativity indicated a lower risk. End-of-treatment (EOT) PET also predicted relapse, but patients with a positive iPET who later achieved a negative EOT PET had worse PFS than those with consistently negative results, highlighting iPET’s role in refining risk assessment. These findings suggest that iPET should be explored as a tool for response-adapted treatment strategies in FL patients.

## Figures and Tables

**Figure 1 cancers-17-01065-f001:**
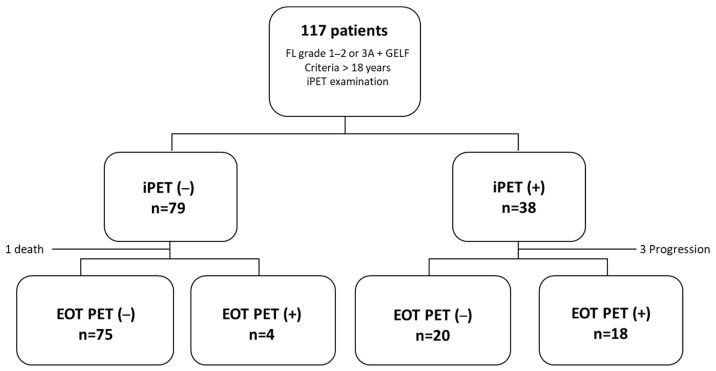
Study profile. We included all patients diagnosed with follicular lymphoma grade 1–2 or 3A, who fulfilled the GELF criteria, were older than 18 years, required treatment, and underwent an iPET examination.

**Figure 2 cancers-17-01065-f002:**
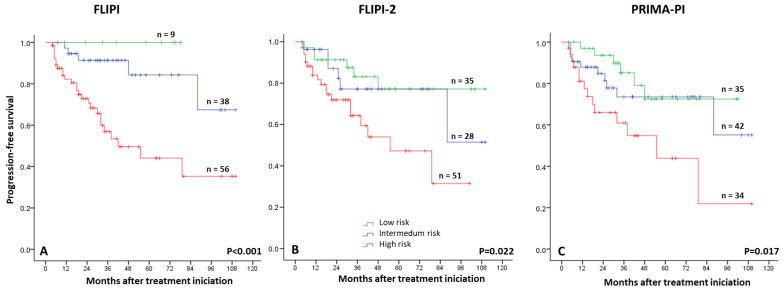
Progression-free survival rate according to risk scores (**A**) FLIPI, (**B**) FLIPI-2, and (**C**) PRIMA-PI.

**Figure 3 cancers-17-01065-f003:**
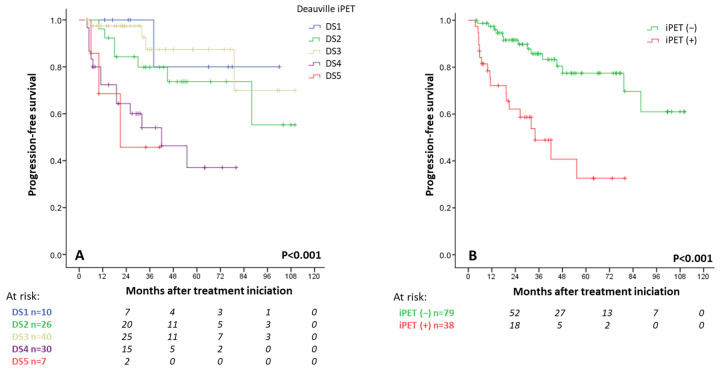
(**A**) Progression-free survival rate according to interim Deauville scale (DS) at interim PET/CT. (**B**) Progression-free survival rate according to interim PET review (threshold DS ≥ 4).

**Figure 4 cancers-17-01065-f004:**
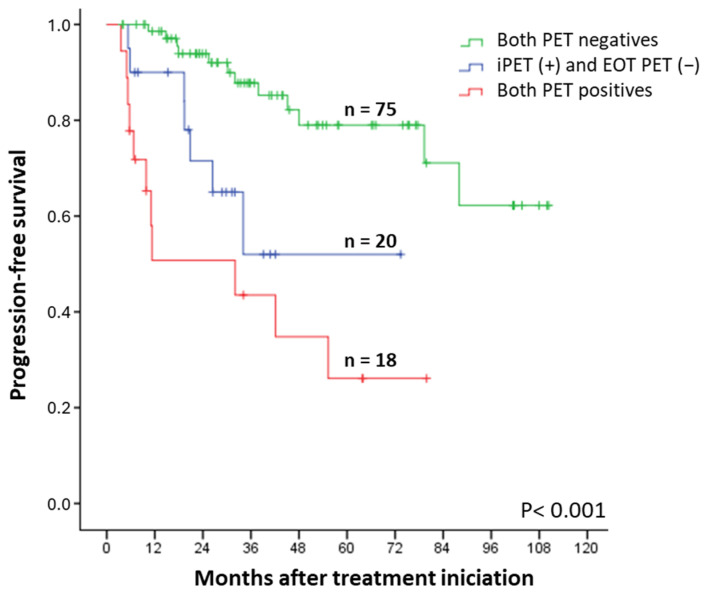
Progression-free survival rate according to interim and EOT PET/CT: both evaluations negative, interim PET (+) and EOT PET (−), or both evaluations positive.

**Table 1 cancers-17-01065-t001:** Patient demographics and baseline characteristics.

Parameter	Total n = 117
Median age, years (range)	62 (27–87)
Sex, male, %	60
Grade, %	
1–2	88
3A	29
Ann Arbor stage, %	
2	9
3	25
4	83
B symptoms, %	23
Bulky disease, %	38
Extranodal involvement, % *	31
Bone marrow involvement, %	57
FLIPI, n = 113, %	
Low risk	9
Intermediate risk	38
High risk	66
FLIPI2, n = 113, %	
Low risk	35
Intermediate risk	27
High risk	51
PRIMA-PI, n = 111, %	
Low risk	35
Intermediate risk	42
High risk	34
Hemoglobin (median g/dL, range)	13.9 (8.8–17.7)
Increased LDH, %	37
Increased β2-microglobulin, %	57
Treatment regimen, %	
R-CHOP	77
Rituximab and bendamustine	21
Other treatments	2
Rituximab maintenance, n = 105, %	87

***** Other than bone marrow.

**Table 2 cancers-17-01065-t002:** Cox multivariate and univariate analyses for progression-free survival rates.

	Univariate *p* Value	Univariate HR (95% CI)	Multivariate *p* Value	Multivariate HR (95% CI)
Age ≥ 60	0.585	1.22 (0.60–2.45)	*	
Female (vs. male)	0.414	0.78 (0.37–1.50)	*	
Grade FL 3A (vs. 1–2)	0.101	0.41 (0.15–1.19)	*	
Advanced Ann Arbor stage	0.310	2.81 (0.38–20.68)	*	
B symptoms	0.929	1.04 (0.47–2.31)	*	
>4 nodal areas	**0.033**	8.74 (1.19–64.03)	0.072	6.58 (0.84–51.3)
BM involvement	0.133	1.76 (0.84–3.76)	*	
Bulky disease	0.778	1.11 (0.53–2.31)	*	
Extranodal involvement ^1^	0.943	1.03 (0.46–2.31)	*	
Elevated β2 microglobulin	**0.002**	3.61 (1.61–8.09)	0.226	1.81 (0.69–4.71)
Hemoglobin < 12	0.109	1.88 (0.89–4.08)	*	
Elevated LDH	**<0.001**	4.17 (1.99–8.78)	0.521	1.38 (0.51–3.73)
RB (vs. RCHOP)	0.655	1.20 (0.54–4.66)	*	
Rituximab maintenance	**0.040**	0.38 (0.15–0.96)	0.055	0.37 (0.13–1.02)

^1^ Other than bone marrow. * Not calculated.

## Data Availability

The datasets generated during and/or analyzed during the current study are available from the corresponding author upon reasonable request.

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
