# Peer review of "Exploratory Study to Evaluate the Impact of Interim PET/CT Assessment in First-Line Follicular Lymphoma"

_cancers, 2025, doi:10.3390/cancers17071065_

Round 1

Reviewer 1 Report

Comments and Suggestions for Authors

You have valuable imaging and clinical data, with a large cohort of patients with FL 1-3A who were evaluated with both iPET and eotPET, but the article as it is written falls short.

The first and most important reason is because the scope of the article is too large, trying to address different questions with very different cohorts (especially for MRD evaluated by cfDNA) in the same article. This should not be done, I think that you should write something separately for cfDNA (probably with more patients). First you should exclude the 4 patients who did not have both iPET and eotPET to get one and only one cohort, to get a still large cohort that is way easier to describe. Secondly, PFS is a good surrogate in this disease that is not deadly, and you should be comfortable with it. HR of 4-5 in terms of OS based on PET response seems good. What your patients allow you to do is a thorough comparison of respective prognostic values of iPET and eotPET on PFS, even if it’s not with statistical tests to compare HR. In my opinion this is what you should discuss. Is the slightly higher HR of eotPET important or not versus the ability to change treatment earlier (in the discussion you can speak about the 3 patients who were refractory to immune-chemotherapy) ? Should one perform both PET or only eotPET or even only IPET in certain cases (when its negative for example) ? For example if you use iPET and FLIPI in your multivariate analyse, what is the risk of relapse within 2 years ? Would it be safe to forego eotPET with DS 1-3 + low flipi ?

When it comes to rituximab maintenance there seems to be enough patients to make so analyses, overall but : 1 / it seems biased as those who didn’t receive rituximab had a good reason for it (like other malignancies or so), so their basal risk of death (and thus of progression according to the classic definition of PFS you used) is higher. 2/ There only a small number of events in the few patients who did not receive it, especially if you further stratify with PET response (only one event in iPET – patients who did not receive maintenance ; clearly not enough to make a log-rank test….)

What seems interesting is how you classify response with DS.. you should give survival curves by DS scores, and compare them to those of Merryman et al, because they classify DS3 as positive and you don’t. Surely this is interesting to discuss.

As for the presentation it would be to long to make a list of things that could be improved. I’ll just take the example of the survival curves with rituximab maintenance (which I think shouldn’t be the focus of this article), would it hurt to add a small “iPET –“ or “iPET+” on each curve ? This would make it easier to read. Think about the reader. He is in a hurry and his attention is limited. Also, the abscissae in months are very different between curves, which is misleading, if you present curves side to side you should use the same for all (easy to edit with SPSS).

Comments on the Quality of English Language

It's not that bad but it could clearly be improved

Author Response

Comments 1: The first and most important reason is because the scope of the article is too large, trying to address different questions with very different cohorts (especially for MRD evaluated by cfDNA) in the same article. This should not be done, I think that you should write something separately for cfDNA (probably with more patients). First you should exclude the 4 patients who did not have both iPET and eotPET to get one and only one cohort, to get a still large cohort that is way easier to describe.

Response 1: We appreciate this comment and agree with the reviewer. To address this, we have removed the section on MRD evaluated by cfDNA. Indeed, we are working on a separate manuscript that includes a much larger patient cohort. Additionally, we have excluded the four patients who did not have both iPET and eotPET, ensuring a single, well-defined cohort.

Comment 2: Secondly, PFS is a good surrogate in this disease that is not deadly, and you should be comfortable with it. HR of 4-5 in terms of OS based on PET response seems good. What your patients allow you to do is a thorough comparison of respective prognostic values of iPET and eotPET on PFS, even if it’s not with statistical tests to compare HR. In my opinion this is what you should discuss. Is the slightly higher HR of eotPET important or not versus the ability to change treatment earlier (in the discussion you can speak about the 3 patients who were refractory to immune-chemotherapy) ? Should one perform both PET or only eotPET or even only IPET in certain cases (when its negative for example) ? For example if you use iPET and FLIPI in your multivariate analyse, what is the risk of relapse within 2 years ? Would it be safe to forego eotPET with DS 1-3 + low flipi ?

Response 2: Thank you for highlighting this important point. We have revised the discussion to clarify this aspect and emphasize the prognostic value of iPET and eotPET on PFS. Additionally, we have included further analysis to support our findings to address the potential clinical implications, including the relevance of the slightly higher HR of eotPET, the timing of treatment modifications, and the role of both PET scans in specific scenarios. Our findings suggest that the slightly higher HR of eotPET is not clinically significant, and these results support the possibility of modifying treatment earlier. This is especially relevant given the recent approval of novel immunotherapy strategies with high efficacy in FL, such as CAR-T therapy and bispecific antibodies. Furthermore, given the excellent outcomes in patients with low-risk FLIPI, we propose that eotPET may not be necessary in cases where iPET is negative. However, further studies are needed to determine whether eotPET can be safely omitted in patients with high-risk PRIMA, FLIPI, or FLIPI-2 scores.

Comment 3: When it comes to rituximab maintenance there seems to be enough patients to make so analyses, overall but : 1 / it seems biased as those who didn’t receive rituximab had a good reason for it (like other malignancies or so), so their basal risk of death (and thus of progression according to the classic definition of PFS you used) is higher. 2/ There only a small number of events in the few patients who did not receive it, especially if you further stratify with PET response (only one event in iPET – patients who did not receive maintenance ; clearly not enough to make a log-rank test….)

Response 3: The reviewer is absolutely right. However, only two patients did not receive rituximab maintenance due to an intercurrent malignancy. In most cases, the reasons were less severe, such as the patient's preference to avoid hospital visits every two months, mild cytopenias, or minor adverse events—mostly flu-like symptoms that did not require hospitalization.Overall, the majority of patients (87%) underwent rituximab maintenance, while 15 did not receive it due to adverse events (n=6), physician decision (n=2), intercurrent malignancy (n=2), or other unknown reasons (n=5).  In any case, we have decided to exclude this sub-analysis and focus on a larger cohort to confirm this hypothesis.

Comment 4: What seems interesting is how you classify response with DS..you should give survival curves by DS scores, and compare them to those of Merryman et al, because they classify DS3 as positive and you don’t. Surely this is interesting to discuss.

Response 4: We completely agree with the reviewer and have included these analyses as Figure 2 in the manuscript. Unlike the study by Merryman et al., our data show that patients with a DS3 score have outcomes similar to those with DS1-DS2. Our findings align with the Lugano Response Criteria (Cheson BD et al., JCO 2024). Further studies are needed to validate whether the results observed by Merryman et al. can be replicated in other cohorts.

Comment 5: As for the presentation it would be to long to make a list of things that could be improved. I’ll just take the example of the survival curves with rituximab maintenance (which I think shouldn’t be the focus of this article), would it hurt to add a small “iPET–“ or “iPET+” on each curve ? This would make it easier to read. Think about the reader. He is in a hurry and his attention is limited. Also, the abscissae in months are very different between curves, which is misleading, if you present curves side to side you should use the same for all (easy to edit with SPSS).

Response 5: We appreciate the reviewer’s feedback. Following this suggestion, we have added “iPET–” and “iPET+” labels to each curve to improve readability. Additionally, we have standardized the time scales across all survival curves to ensure consistency and avoid misinterpretation. Thank you for highlighting these important points to enhance clarity for the reader.

Reviewer 2 Report

Comments and Suggestions for Authors

The authors present an investigation into interim PET scans conducted during treatment, including liquid biopsy, perhaps to predict outcomes for patients with FL undergoing their first treatment. 

The topic is of interest and appears to be well-written.

Comments:

1) The title of the small study should include "Exploratory Study" or "Proof of Concept" to alert the readership that it is a small study that adds to the literature but is still relatively an area of investigation.

2) One interesting aspect was the liquid biopsy, but it appears to have low counts that influence patient pathways, according to the article, and should be clarified in the text. Also, the cost of the liquid biopsy should be mentioned (or approximated). Contigency table needed as discussed further below.

3) Was Baseline status available from the patient (such as from bone marrow biopsy?) and/or is there any subsequent information on pathology reads?  Is there any information on anemia status?

4) The authors did an extensive write-up with many details of groupings of patients, and that is of high value in this paper for the readership to understand this data. Would it be possible to submit the images to a repository of some form to go along with the other data so others can expand on this information in the future (that would create more greater value for future research)?

5) Please consider contrasting your protocols to both EMSO, NCCN and ASH protocols that may overlap with your protocol (How do the authors protocol fall within guidelines and cite that)

6) While this is a clinical study, it is always useful to discuss more control over the calibration. Was there calibration performed for the machine (nuclear/medical physics, when was that done?, perhaps useful if it was done near the time at the onset of the study for future or current reference).  was it tested after the study, or during. Add to discussion or provide data.

7) What is the 'protocol' for selecting the max SUV (is there guidance on how that is selected so others can follow a process) that is similar to yours? For example, would you rule out areas not to take the max values from as that could be artifactual?  Also, is there consideration or report for Body weight, surface and or body mass index. Add to methods.  Suggestion: Consider if there are specific nuclear readers  their 'initials' could be referenced as they are 'experts' and of utility for others to know that they signed off on this part of the study (particularly if they are authors on this paper, that wasnt clear in the text, they could also be in acknowledgements - they are adding value to the study and expertise)

8) section 3.8.  A contingency table with axes that include cfDNA analysis and iPETct results as well as potentially outcomes described would help provide better information to users.  That would indicate the 'scale' of the predictive value beyond ratios. 

9) In discussion, it is valuable to describe the balance and availability for equitable access to PET/CT and to liquid biopsy.  How is that managed in your area?

10) MRD seems to be used in conjunction with different references (attached) to the ligBio as well as MRD alone.  For the reader's purpose, a definition and understanding of how minimal residual disease (MRD) is thought of by the authors would help readers evaluate/understand the concept of how it is used in the clinical sense for this project. Which parts are liquid, which parts are pathology, which parts are NGS, which parts may be iPET for example if relevant.

Author Response

Comment 1: The title of the small study should include "Exploratory Study" or "Proof of Concept" to alert the readership that it is a small study that adds to the literature but is still relatively an area of investigation.

Response 1: We appreciate the reviewer’s suggestion. We have modified the title to "Exploratory Study to Evaluate the Impact of Interim PET/CT Determination in First-Line Follicular Lymphoma."

Comment2: One interesting aspect was the liquid biopsy, but it appears to have low counts that influence patient pathways, according to the article, and should be clarified in the text. Also, the cost of the liquid biopsy should be mentioned (or approximated). Contigency table needed as discussedfurtherbelow.

Response 2: Since this point has been raised by both reviewers, and we are currently working on a new cohort of 150 patients treated in the first line with cfDNA determination in liquid biopsy, we have decided to remove these preliminary analyses from the present manuscript.

Comment 3: Was Baseline status available from the patient (such as from bone marrow biopsy?) and/or is there any subsequent information on pathology reads?  Isthereanyinformationon anemia status?

Response 3: Thank you for your insightful comment. As described in the manuscript, we have included a comprehensive set of baseline variables at diagnosis, including bone marrow infiltration and anemia status, which can be found in Table 1. Additionally, for patients with bone marrow infiltration at diagnosis, a reassessment was performed at the end of treatment. To be considered in complete remission (CR), the bone marrow had to be free of infiltration.

Comment 4: The authors did an extensive write-up with many details of groupings of patients, and that is of high value in this paper for the readership to understand this data. Would it be possible to submit the images to a repository of some form to go along with the other data so others can expand on this information in the future (that would create more greater value for future research)?

Response 4: We appreciate the reviewer’s suggestion. All images will be available to readers upon request to ensure accessibility and support future research.

Comment 5: Please consider contrasting your protocols to both EMSO, NCCN and ASH protocols that may overlap with your protocol (How do the authors protocol fall within guidelines and cite that)

Response: We appreciate the reviewer’s suggestion. When discussing the treatment received by our patients, we refer to the GELTAMO guidelines (ISBN: 978-84-09-44428-1), which provide specific recommendations relevant to our study.

Comment 6: While this is a clinical study, it is always useful to discuss more control over the calibration. Was there calibration performed for the machine (nuclear/medical physics, when was that done?, perhaps useful if it was done near the time at the onset of the study for future or current reference).  was it tested after the study, or during. Add to discussion or provide data

Response 6: Thank you for your suggestion. The calibration of the machines was performed annually at Hospital 12 de Octubre, and they undergo yearly AERL accreditation by the European Association of Nuclear Medicine (EANM). This information has now been included in the methodology section.

Comment 7: What is the 'protocol' for selecting the max SUV (is there guidance on how that is selected so others can follow a process) that is similar to yours? For example, would you rule out areas not to take the max values from as that could be artifactual?  Also, is there consideration or report for Body weight, surface and or body mass index. Add to methods.  Suggestion: Consider if there are specific nuclear readers  their 'initials' could be referenced as they are 'experts' and of utility for others to know that they signed off on this part of the study (particularly if they are authors on this paper, that wasnt clear in the text, they could also be in acknowledgements - they are adding value to the study and expertise)

Response 7: Thank you for your valuable comment. The SUVmax was calculated using the following formula:

                      Activity concentration in the lesion (Bq/mL)

SUVmax =

                              Injected dose (Bq) / Body Weight (kg)​

Regarding the selection protocol for SUVmax, we followed standard guidelines to minimize artifacts. Areas prone to physiological uptake or potential artifacts were carefully reviewed and excluded by Pilar Sarandeses, nuclear medicine expert and coauthor of the study.

Comment 8: A contingency table with axes that include cfDNA analysis and iPETct results as well as potentially outcomes described would help provide better information to users.  That would indicate the 'scale' of the predictive value beyond ratios.

Response 8: As previously mentioned, the cfDNA analyses have been removed from this manuscript to improve clarity and readability. These analyses will be published as part of a larger study.

Comment 9: In discussion, it is valuable to describe the balance and availability for equitable access to PET/CT and to liquid biopsy.  How is that managed in your area?

Response 9: Liquid biopsy is still an experimental technique within the scope of research and is not yet used in real-life clinical practice.

Comment 10: MRD seems to be used in conjunction with different references (attached) to the ligBio as well as MRD alone.  For the reader's purpose, a definition and understanding of how minimal residual disease (MRD) is thought of by the authors would help readers evaluate/understand the concept of how it is used in the clinical sense for this project. Which parts are liquid, which parts are pathology, which parts are NGS, which parts may be iPET for example if relevant.

Response 10: We have decided to remove MRD analyses from this manuscript to improve clarity and focus.

Round 2

Reviewer 1 Report

Comments and Suggestions for Authors

The current version is much clearer in my opinion, which does more justice to this interesting article. Please verify typo, especially around lines 266-277(“os”, sentence repeated twice…), as there are many of them in this area of the text. 

Comments on the Quality of English Language

English could be improved but the article is understable as it is. Ex: should be taken with care and not “it should take with care”

Reviewer 2 Report

Comments and Suggestions for Authors

article improved